# Systematic Literature Review of Capitation and Fee-for-Service Payment Models for Oral Health Services: An Australian Perspective

**DOI:** 10.3390/healthcare9091129

**Published:** 2021-08-30

**Authors:** Jennifer H. Conquest, Nirjgot Gill, Praveena Sivanujan, John Skinner, Estie Kruger, Marc Tennant

**Affiliations:** 1School of Human Sciences, University of Western Australia, Perth 6009, Australia; jhconquest@gmail.com (J.H.C.); estie.kruger@uwa.edu.au (E.K.); marc.tennat@uwa.edu.au (M.T.); 2Dentistry, James Cook University, Cairns 4870, Australia; nirjyot.gill@my.jcu.edu.au (N.G.); praveena.sivanujan@my.jcu.edu.au (P.S.); 3Faculty of Medicine and Health, University of Sydney, Sydney 2006, Australia

**Keywords:** dental, capitation, fee-for-service

## Abstract

The aim of this review was to assess relevant global literature on capped-fee (CF) and fee-for-service (FFS) payment models as used by public dental services. Research data were assessed through the PRISMA check list and sourced from MEDLINE, PubMed, ProQuest, Cochrane Library, and other methods. The inclusion criteria were peer reviewed articles published between 2004 and 2020 and (i) other countries’ health systems that were evaluated in contrast to Australia; (ii) care provided to individuals; (iii) payment models for private services that were the same as Australian government policy (CF and FFS); and (iv) care provided by dentists. We used a mixed methodology for data collection. A total of 262 references were reviewed with 10 references meeting the inclusion criteria with the quality rating being: three—strong, six—moderate, and one—weak. The literature included studies from Sweden (three references), Ireland (three references), United Kingdom (six references), United States of America (two references), and Norway (one reference). Four references included studies within multiple countries. The sample size varied between 20 and 106,874 participants. The two payment systems can impact on individual outcomes, such as by overtreatment in an FFS system and undertreatment in a CF system.

## 1. Introduction

The implementation of cost containment tools dates to 1970 but its universal use in Organization for Economic Cooperation and Development (OECD) countries has only been apparent since 1990 [1]. The need for cost containment arose when the ability to pay became restricted by budget constraints that increased the costs of the population’s health needs [1]. Containing costs can be a useful tool for governments that provide public health care, but it may have a negative impact on innovation, quality of care, and profit margins for the private sector [2]. Both fee-for-service (FFS) and capped-fee/capitation (CF) are utilised in outpatient health and hospital settings; however, in the United States of America (USA) CF is used in both outpatient and inpatient settings [1]. CF is the preferred containment tool for providing services to high-risk population groups that contain adjustments for age and sex [1].

Global discussions about priority settings have focused on addressing efficiency and equity, targeting marginalised populations, and integrating oral health into other general medical services [3,4]. These debates target access and treatment processes as well as payment systems regarding the private dental sector [5]. Worldwide analysis has been undertaken in relation to the burden of disease as identified in epidemiological patterns of communicable and non-communicable disease trends [6]. Within the ever-changing political arena, an ageing population, epidemic disease impacts, and associated social and economic repercussions in global society, the topic of payment systems engaging private practitioners to offer public health care remains contemporary.

Currently the most successful OECD countries in implementing FFS are France, Germany, and Japan for medical practices [7]. Payment prices for healthcare are an important component in maintaining universal access to care, like Australia’s Medicare system, so it is important to find the right balance between payment to providers and costs to the government and the taxpayer.

The aim of this systematic literature review is to assess the relevant literature on payment models for public dental services focused on FFS and CF that are designed to target dental practitioners/facilities and specific consumer high-risk target cohorts. Thus, the analysis required multiple aspects to be considered: (i) cost–benefit analysis; (ii) practitioner/facility participation and views on pricing; (iii) patient views on participation and quality of care; and (iv) relevance to Australia’s government policy on utilising private services to deliver public dental care.

## 2. Materials and Methods

The study used the PRISMA method as the basis for assessing the review findings [8] and the PRISMA checklist of items for reporting was followed.

The EPHPP Global rating tool was evaluated by Thomas [9] to assess its successful adaptability to contemporary methods of systematic reviews for investigating questions relating to public health care. This tool demonstrated its ability to appraise research that ranged from health promotion interventions and random clinical trials to non-randomised studies [9]. Additionally, the EPHPP tool also demonstrated its effectiveness in a systematic review that incorporated studies based on different designs and durations. For example, Chillón [10] conducted a systematic review that included study designs of quasi-experimental, observational, randomized control trials, with duration times of 4 weeks to 4 years.

We restricted publication time to the last 16 years as literature debating FFS and CF goes back to 1990. We decided to focus on more recent papers (2004–2020) to ensure that the systematic literature review would be relevant to today’s issues of procuring private practitioners/facilities to provide public dental care.

The systematic process applied the PICO Framework [11] to obtain the best possible outcome for providing public dental care. The targeted cohort was low socioeconomic adults over 18 years of age who accessed public dental services. For public dental services to meet demand requires more private dental practitioners/facilities. We compared: (i) the contractual agreements applying different payment systems methods (FFS and CF); and (ii) the associated impact on the private dental practitioner/facility and the individual who received care. The desired outcome is for the systematic literature review to provide practical guidance for policy makers on the strengths and weaknesses of both FFS and CF in addressing oral health service needs.

### 2.1. Search Strategy and Eligibility Criteria

This systematic review searched 4 electronic databases, websites, and organisations using key terms and citation searching. The databases were MEDLINE, PubMed, ProQuest, and Cochrane Library. The inclusion criteria were: (i) other countries’ health systems evaluated in contrast to Australia; (ii) care provided to individuals; (iii) payment models for private services that were the same as Australian government policy (CF and FFS); (iv) care provided by dentists. We used a mixed methodology for data collection, which included only those titles published in the last 16 years (2004–2020). Only English language papers were included because of cost, time, funding, and language resource constraints.

The key terms used were capitation (CF), procured services, dental, dental services, oral health, payment types, practitioner type, practitioner options of CF, and patient options of CF.

There were no restrictions on the clinical setting or location of the study. However, “grey” literature from government and non-government organisations was excluded, as it consisted of reports, strategic planning, policy statements, and discussion/issue papers.

### 2.2. Data Extraction

The general characteristics of each article including participants, details of the payment model(s) being tested, along with the study measures and outcomes were extracted.

### 2.3. Quality Assessment

The references that met the inclusion criteria were reviewed with the Effective Public Health Practice Project (EPHPP) Quality Assessment Tool for Quantitative Studies and allocated a global quality rating [9]. This included 8 quality assessment domains (selection bias, study design, confounders, blinding, data collection methods, withdrawals and dropouts, intervention integrity, and analysis). The EPHPP weightings are such that if an article is assessed as ‘strong’ it can have no weak ratings, ‘moderate’ can have 1 weak rating, and ‘weak’ can have 2 or more weak ratings. In relation to selection bias, ‘strong’ is assessed if the selected individuals for the study are highly likely to be representative of the target population and there is greater than 80% participation.

The EPHPP Dictionary was used to clarify the objective of each of the assessment domains.

## 3. Results

The total number of references that were assessed for eligibility was 15 (Figure 1). However, only 10 references met the criteria for quality assessment.

### 3.1. Reference Selection

A total of 237 records were identified from the 4 databases and 25 records via websites, organisations, and citation searching. Two duplicates were removed, and the remainder assessed by their titles and abstracts. The exclusion criteria were: publications older than 2004, topics not relevant to the aim of the paper, articles without sufficient information regarding methodology and not able to be assessed with the EPHPP tool, and data not from a recognised source. This resulted in 141 selected with a further 131 removed upon reading the references in full. Therefore, a total of 10 (eight studies and two reports) passed the EPHPP assessment (Figure 1). Data were extracted from the 10 references and are displayed in Table 1.

### 3.2. Study Design

Reference designs included a quasi-experimental study [12], a longitudinal study [13], four systematic reviews [14,15,16,17], two qualitative research studies [18,19], a narrative literature review [20], and a cross-sectional study [21].

There were variances within the study designs. The systematic review by Brocklehurst et al. [14] contained three streams of methodology: (i) quantitative, (ii) qualitative, and (iii) questionnaire, while the Woods review [15] contained randomized controlled trials. The narrative literature review of Voinea-Griffith et al. [19] included analyses of dental care, evidence-based dentistry, outcome indicators, and diagnostic codes, while the cross-sectional study of Whittaker and Birch [21] used longitudinal data from the 1991–2008 waves of the British Household Panel Survey.

### 3.3. Quality Assessment

Using the EPHPP Global rating, the 10 references were rated as either strong, moderate, or weak. Three were classified as strong, seven were moderate and one weak (Table 1). Two of the studies were rated strong for selection bias, suggesting that the selected individuals were highly likely to be representative of the target population and there was greater than 80% participation (EPHPP Dictionary). Seven of the remaining studies were rated as weak for selection bias. This could be due to various reasons such as the selected individuals not likely to be representative of the target population or less than 60% participation. Alternatively, selection bias can be weak if the selection and the level of participation were not described (EPHPP Dictionary).

It was challenging to assess bias due to the different types of methodology, e.g., systemic review. This risk of bias was particularly evident in and due to the quasi-experimental study design. References with notable risk of bias are discussed below.

Andäs and Hakeberg’s article (2014) [12] acknowledged that because of the nature of the quasi-experimental design, there were selection bias consequences that included overinterpreting the results. However, this was counterbalanced by a balanced sex representation (47.3% males, 52.7% females).

In comparison, the Andäs and Hakeberg article (2016) [13] found two areas of possible bias. The study was a quasi-experimental design and used a large variety of data collectors because of the many centres. This multi-centre arrangement had the potential to result in the possible misclassification of caries, whereas in Brocklehurst et al. [14] there was no certainty that selection bias was not present.

In references where questionnaires were used, a response bias may be expected. For example, responses are more likely to come from more enthusiastic and passionate participants and practitioners. This risk cannot be accounted for or measured accurately. In Woods [15], questionnaires/surveys were used to collect participant data. Participant selection bias was prevalent in the above study as the individuals were selected from a pool of rural patients. These individuals may have been more disadvantaged as they may not have access to fluoridated water or to dental services and may have had a higher prevalence of dental disease. To improve the selection bias and issues of undertreatment of participants, adjustment to the per capita fee can be made. Differentiation of the per capita fee can be made through identifying different patient groups and treating them as needed (i.e., patients with a high incidence of caries can be identified and treated with a different per capita fee).

### 3.4. Study Population

The 10 references included studies within the following counties: Sweden (three refences), Ireland (three references), UK (six references), USA (two references), and Norway (one reference). Four references included studies within multiple countries (Table 1). The sample sizes varied between 20 and 106,874 participants.

In the studies where sex was reported (*n* = 2) there was an equal proportion of male and female participants. All 10 references covered both rural and urban areas in their analysis. The adult age groups in some references were presented in groupings. In the Andäs and Hakeberg [12,13] studies their adult cohort was 20 years of age and older. In the Brocklehurst et al. [14] and Strand et al. [18] studies, the adults were ≥24 to 60 years of age, while Whittaker and Birch [21] and Woods [15] studied adults of 16 years and older (Figure 2).

### 3.5. Dental Provider

Woods [15], Whittaker and Birch [21], Voinea-Griffith et al. [16,19], Strand [18], Hill [17], and Brocklehurst [14] studies focused on contracted private dental practitioners. Tickle [20], Andäs [12], and Whittaker and Birch [21] analyses included public dental practitioners. The study of Strand [18] also included examination of a private dental organisation (Figure 3).

### 3.6. Interventions

In three of the selected references [12,13,21], all participants received the allocated intervention or exposure of interest, described as a payment program. In all remaining references the intervention was not applicable to the study design, as participants were patients who were required to adhere to a payment scheme and did not have a choice to decline. In 7 out of 10 references, the intervention was constant and measurable in the form of qualitative data (through answers from patient questionnaires). In the remaining selected references [19,20], the level of intervention could not be measured.

In four selected references [13,14,18,20] the subjects did not receive unintended intervention (contamination or co-intervention). The EPHPP definition of “contamination or co-intervention” is where the “control group” unintentionally obtains the research study’s intervention. This can result in an underestimation of the impact of the intervention. For the remaining studies, contamination was not measured and stated. All the references included the same intervention strategies of targeting practices/offices. Vionea-Griffith et al. [19] targeted an organisation/institution. All studies were analysed on the actual intervention rather than on the diagnosis of what treatment was required, and all studies focused on data from a cohort of individuals.

In the research conducted, blinding and withdrawal rates were not relevant because participants were required to stay engaged with one type of provider plan for the duration of their treatment at their chosen practice. Both assessors (e.g., practitioners and practice owners) and participants were aware of the data collection methods (questionnaires) and the interventions that were measured. However, the research question was not revealed to the participants in any of the references.

### 3.7. Outcomes

Four out of the 10 selected references suggested that the participants favoured CF. Those who selected CF reported being healthier and more engaged in health-promoting behaviours. The study by Andäs and Hakeberg [12] noted that females who were well-educated and made good lifestyle choices and those who acknowledged the importance of good oral health [12] were more likely to participate in CF. One of the studies [17] reported that the move to CF from FFS suppressed clinical activity, including prevention. However, selected references [12,13] concluded there is a higher health risk of carious lesions in the FFS model of care.

## 4. Discussion

The current study reviewed the existing evidence regarding the complexities of the payment systems (CF and FFS) used in the private sector to support the service delivery for dental care.

Schneider et al. [22] reported on the differences in health systems and placed them in the following order of ranking: (1) UK; (2) Australia; (3) Netherlands; (4) New Zealand and Norway; (6) Switzerland and Sweden; (8) Germany; (9) Canada; (10) France; and (11) USA. The top three countries (United Kingdom, Australia, and Netherlands) obtained this ranking by providing universal coverage and access through utilising various payment systems. Universal coverage is described by three systems: Beveridge, Single payer, and Multiplayer. The UK uses the Beveridge system, which sources money from the general tax revenue. This has direct accountability to the government. Australia has a Single payer system that uses universal health insurance under a public insurance plan, funded through general tax revenue; however, approximately half the population purchases private health insurance. The Netherlands has a Multiplayer system that relies on private insurance to provide care to the population, financed through community-rated premiums and payroll taxes [22].

In Australia 87% of dental services are provided privately [23]. The difference between working as a private or a public dentist is that a public dentist is paid a fixed salary and is restricted to providing dental care as determined by state and federal governments, whilst a private dentist is paid by FFS by a health insurance company, the individual, or a combination of both. Dental care for the private paying individual is not restricted by state/federal policies. To address the demands of equity (access to services and quality of care process) to public health services, where Australia ranks 7th in the world, outsourcing to private providers is an important population strategy [22]. Public health services throughout Australia use various models of FFS and CF [24]. It is therefore important that the limitations of both payment models for procuring care are understood. For example, in dental care it has been claimed that FFS leads to overtreatment, which might explain to some extent the rise in caries rates of 1.5, while CF might lead to adverse selection and undertreatment [12,13,25]. Patients feel more engaged in a CF approach as it has a greater preventive focus [12].

Important arguments were raised within the selected references to improve the payment systems, such as better communication concerning the contract and risk assessment [18]. Additionally, based on Voinea-Griffith et al.’s [19] study on evidence-based dentistry and United States dental practice patterns, implementing pay for performance for dental care appears to be premature. Moreover, the findings suggested that changing the way dentists are paid on its own is unlikely to achieve all policy goals. Therefore, research is required to develop and test supplementary interventions that can work in concert with a remuneration system to achieve desired policy outcomes [17]. In addition, future oral health policy will need to address oral health inequalities, encourage skill mix, and promote and facilitate dental professionals to deliver appropriate and high-quality care relevant to the needs of their local population [26].

### Limitations

A limitation of this systematic review is in comparing the different cohorts within the selected references as they often did not analyse the same criteria such as age groups, e.g., adults starting at 16 years or at 20 years of age. This was also the case with various aspects of treatment within the payment schemes. However, this did not weaken the findings that there is a variance in quality of care between CF and FFS and that public health services will remain dependent on the private dentist to help meet demand.

## 5. Conclusions

The two payment systems (FFS and CF) can have impacts on individual outcomes, such as overtreatment in an FFS system and undertreatment in a CF system. Population health strategies targeting marginalised populations need a consistent approach to address oral health inequalities, encourage skill mix, and promote and facilitate the dental professionals to deliver appropriate and high-quality care irrespective of the payment system.

## Figures and Tables

**Figure 1 healthcare-09-01129-f001:**
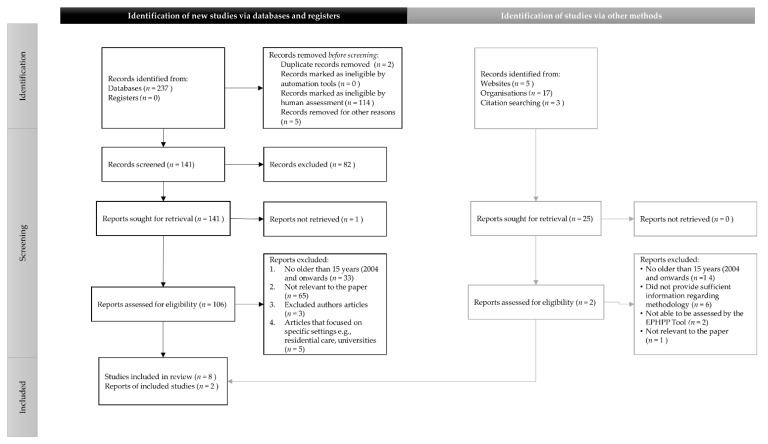
PRISMA flow diagram for reference inclusion.

**Figure 2 healthcare-09-01129-f002:**
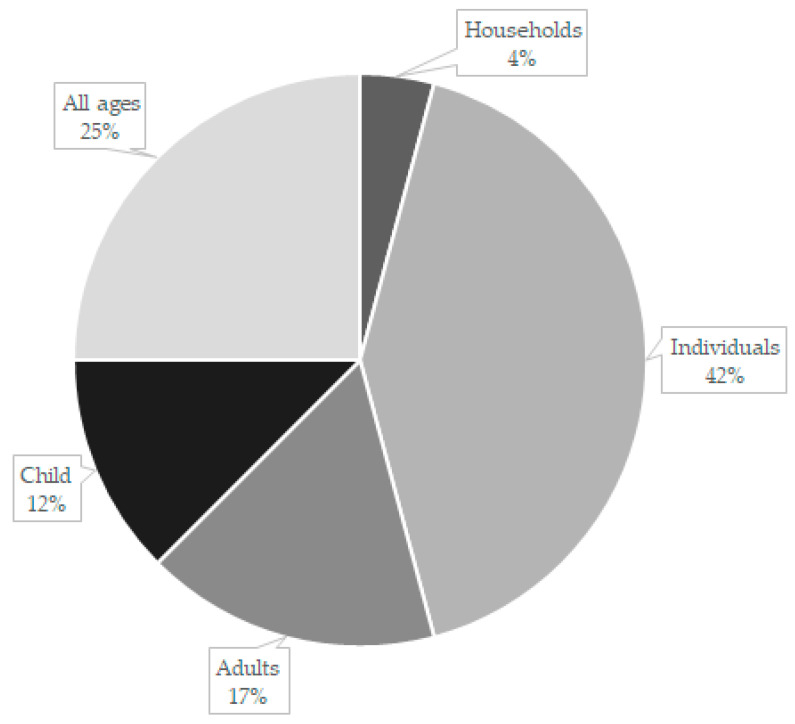
Cohort demographics.

**Figure 3 healthcare-09-01129-f003:**
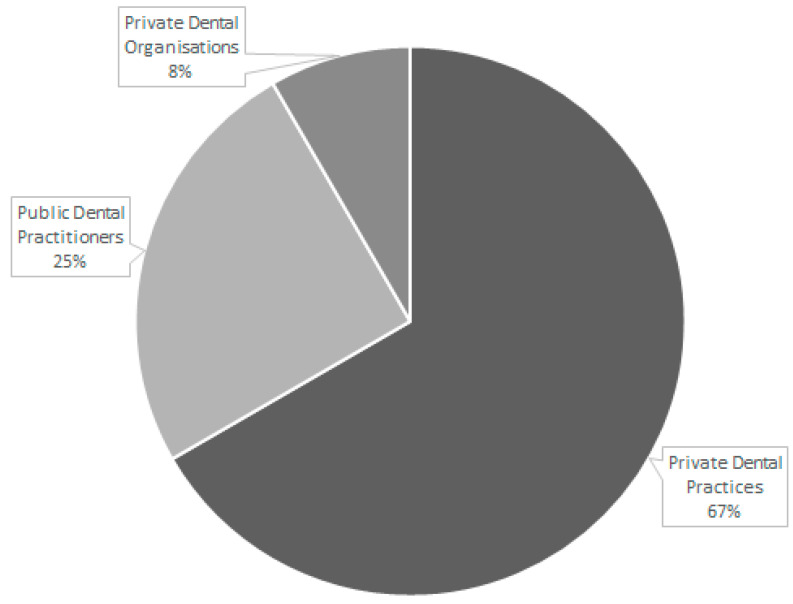
Dental provider demographics.

**Table 1 healthcare-09-01129-t001:** Summary of selected references.

Study (*n* = 10)	Title	Objective	Study Design	Selection Bias and Cohort, Population	Outcome(s)	EPHPP Global Rating
Andäs C A., Hakeberg M (2014) Sweden [12]	Who chooses prepaid dental care? A baseline report of a prospective observational study	Describes potential differences regarding socioeconomic and lifestyle factors, perceived oral health and attitudes towards oral health between patients in the two payment systems (Swedish capitation system and the FFS within the Public Dental Service).	Moderate quasi-experimental design	Strong 13,719 patients enrolled from 20 PDS clinics selected through a stratified, random procedure. Participants who were ≥20 years (male and female) chose their service type. Participants had education levels.	Patients who chose to prepay differed statistically significantly (e.g., BMI, sex, and age) than those who chose to pay traditionally.	Strong
Andäs C A., Hakeberg M (2016) Sweden [13]	Payment Systems and oral health in Swedish dental care: observations over 6 years	The aim of this study was patients in regular dental care compare to the findings of manifest caries and fillings after a 6-year period of care. Either through FFS or capped-fee, e.g., Dental Care for Health (DCH).	Weak longitudinal study	Moderate The target population was 485,000 number of adults ≥20 years (male and female) was 6229. There was 100% participation rate.	The incidence rate ratio of manifest caries lesions after six years in FFS was 1.5 times higher than capped-fee (DCH).	Moderate
Brocklehurst P, et al., (2020) Northern Ireland [14]	Impact of changing provider remuneration on NHS general dental practitioner services in Northern Ireland: a mixed-methods study	Study looked at the levels of care between capped-fee and FFS back to capped-fee.	Strong systematic review	Weak Control practices, practices varied in practice size. Behaviours of equity-owning practice principles and non-equity-owning associate dentists. Age groups were that of children and ≥24 to 60 years of age. The participants were not representative of the target population.	A move from FFS to capped-fee had little impact on access but produced large reductions in clinical activity and patient charge income. Patients noticed little difference in the service that they received.	Moderate
Woods, N. (2013) United Kingdom, Norway, and Ireland [15]	The role of payments systems in influencing oral health care provision	Among the leading strategies to reform health care is the development and implementation of new payment models. The goal is to change the way physicians, dentists, hospitals, and other care providers are paid to emphasise value for money.	Strong systematic review	Weak Participants were children and adults (aged 16 years). Unable to determine participant rate.	The optimal dental contract may be a ‘blended’ payment system whereby dentists receive a proportion of their income through capped-fee, a proportion from allowances and proportion from FFS.	Moderate
Voinea-Griffith A, et al., (2010b) United Kingdom and United States of America [16]	Pay for performance: will dentistry follow?	This article explored factors that would influence the adoption of value-based purchasing programs in dentistry.	Strong systematic review	Weak Analysis covers such aspects as variation in dental care, evidence-based dentistry, outcome indicators, diagnostic codes. Cohort covers all ages; age groups and sex not identified. Unable to determine participant rate.	Discussion on dentists’ performance under FSS and who work in the community and/or dental organisations.	Moderate
Hill H, et al., (2020) Northern Ireland [17]	The impact of changing provider remuneration a clinical activity and quality of care: Evaluation of a pilot NHS contract in Northern Ireland	This study was a pilot to introduce NHS general dental practitioner contractual system in Northern Ireland (2015 and 2016).	Moderate difference-in-difference (DiD) evaluation	Weak Participants were a mix of socioeconomic status, both child and adult and no sex or age groups identified. There was less than 60% of the cohort agreed to participate.	Overall, the move to a capped-fee from FFS suppressed clinical activity, including prevention.	Moderate
Strand J, et al., (2015) Sweden [18]	A new capitation payment system in dentistry: the patients’ perspective	Explore patients’ experiences and attitudes to a new dental payment system regarding the contract, risk assessment, dental care content and economy, as well as the advantages and disadvantages of the payment system.	Weak qualitative research	Weak 20 interviews with 12 women and 8 men between the ages of 24 to 60 years. 77% of the cohort agreed to participate.	Patients were generally in favour of the capped-fee.	Weak
Voinea-Griffith A et al., (2010a) United Kingdom and United States of America [19]	Pay for performance in dentistry: what we know	Current experience of Pay for Performance system (P4P-FFS) in primary medical care that has relevance to dentistry and discuss the dental performance-based programs to date.	Strong qualitative analysis	Weak Cohort covers all ages; age groups and sex not identified. Unable to determine participant rate.	FFS could be successful if there is: (i) explanation of knowledge; (ii) increase in evidence-based clinical guidelines; and (iii) evidence-based performance measures tied to existing clinical practice guidelines.	Moderate
Tickle M. (2012) United Kingdom [20]	Revolution in the provision of dental services in the UK	This paper provides a historical overview of NHS dental services and some personal reflections on the main challenges over the next five years.	Moderate study on narrative literature review (not systematic) and some subjective comments included	Weak Article is based on other articles analysing pilots of capped-fee and FFS that is paid for NHS budget. Participants (child and adult) and age or sex were not identified. Unable to determine participant rate.	Article discusses the history of payment (capped-fee and FFS) transitions of the NHS between 2205 and 2010 as well as historical dental services (1948). It concludes that the NHS will be in financial trouble in years to come because of the inability to control its finances.	Moderate
Whittaker W., Birch S. (2012) United Kingdom [21]	Provider incentives and access to dental care: Evaluating NHS reforms in England	England aimed at improving access to populations with low use. This included: (i) commissioning of NHS dental services by primary care trusts (ii) replacing FFS patient charges by capped-fees and (iii) changing the remuneration of dentists providing dental care.	Strong longitudinal study	Strong 5000 households, and approximately 10,000 individuals (≥16 years and sex not identified).	Evidence shows a decrease in NHS use, driven by those who had previously good access to care. This trend had positive effects on the consumer’s transitions from NHS to private practice. This transition relied upon the ability of the private sector to absorb the demand.	Strong

## Data Availability

Not applicable.

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
