# Peer review of "Systematic Literature Review of Capitation and Fee-for-Service Payment Models for Oral Health Services: An Australian Perspective"

_healthcare, 2021, doi:10.3390/healthcare9091129_

Round 1

Reviewer 1 Report

That´s an excellent manuscript, in a theme with great importance in health area

Author Response

Reviewer 1 . Thank you for your positive feedback.

Reviewer 2 Report

I'm not sure I understand the purpose of the study .

What are you trying to summarize , what is the outcome you want to measure and then compare? Is from the patient point of view, the payment to the dentist? Whose preference ?

Because of the inclusion criteria and mixing different research designs the study is most correctly a literature review and not a systematic review.  This is also supported by your tables and the different outcomes you collected.

Author Response

The responses are summarized in the attached document.

Reviewer 3 Report

Abstract: Please avoid the title of each paragraph: background, ..... Line 12: the choice of 2004 to 2020, why? Line:18: What is 125?   Introduction: This part is very short please    Results: Figure 1: please enlarge the letters in the last column Table 1: please add the number of each reference Figure 2: please use the same typing mode which was used in Figure 1 Line 187: please change: the top three countries (1, 2 and 3) or mention the countries names Line 229: (2) ? Line 233: . to ,   References: please modify according to MDPI style There is two numbers for each reference, please modify Line 276: 00 ??

Author Response

(The authors gave the same response as above.)

Reviewer 4 Report

Manuscript submitted by Skinner and co-authors “Systematic review of capitation and fee-for-service models for oral health services: an Australian perspective”. presents a topic that is interesting. However, there are some aspects that the authors should reconsider.

Introduction: it is very little, lengthen it.

Material and method: there is a new PRISMA protocol (PRISMA 2020) https://pubmed.ncbi.nlm.nih.gov/33782057/ update it.

PICO question?

Table 1 puts n = 11 when everywhere you say it is 10 (results, flow diagram, reference selection, through the manuscript...)

Why do you use the EPHPP to evaluate all the selected articles if each one is of a type and there are specific questionnaires for each one of them?

Sentence 106, why put (7) behind the word seven?

Figure 2, why mix in the same figure ages and type of dentist? Divide it in two.

Sentence 171, delete point and space in front of Outcomes.

Author Response

(The authors gave the same response as above.)

Round 2

Reviewer 2 Report

This is a much improved version.

Your PICO will benefit from a more clear outcome/outcomes. 

Author Response

Thanks for your suggested changes. Our responses are in the attached document.

Reviewer 4 Report

In the entire document n = 10, however, in the summary of articles there are 11. If this is the case, why has one more been selected? or why do you have to remove one more? What reasons does that have for being wrongly selected?

material and methods there are different letters / sizes, unify

It should be put in the flowchart, the 25 records identified through other methods, which "other methods" were?

I do not see the meaning of this systematic review, nor the usefulness

Author Response

Thank you for your comments and suggested changes. These have been addressed in the attached document.
